# Enhanced Li-Ion Rate Capability and Stable Efficiency Enabled by MoSe_2_ Nanosheets in Polymer-Derived Silicon Oxycarbide Fiber Electrodes

**DOI:** 10.3390/nano12030553

**Published:** 2022-02-06

**Authors:** Sonjoy Dey, Shakir Bin Mujib, Gurpreet Singh

**Affiliations:** Department of Mechanical and Nuclear Engineering, Kansas State University, Manhattan, KS 66506, USA; gurpreet@ksu.edu

**Keywords:** anode, battery, ceramic, layered materials, TMD, MoSe_2_

## Abstract

Transition metal dichalcogenides (TMDs) such as MoSe_2_ have continued to generate interest in the engineering community because of their unique layered morphology—the strong in-plane chemical bonding between transition metal atoms sandwiched between two chalcogen atoms and the weak physical attraction between adjacent TMD layers provides them with not only chemical versatility but also a range of electronic, optical, and chemical properties that can be unlocked upon exfoliation into individual TMD layers. Such a layered morphology is particularly suitable for ion intercalation as well as for conversion chemistry with alkali metal ions for electrochemical energy storage applications. Nonetheless, host of issues including fast capacity decay arising due to volume changes and from TMD’s degradation reaction with electrolyte at low discharge potentials have restricted use in commercial batteries. One approach to overcome barriers associated with TMDs’ chemical stability functionalization of TMD surfaces by chemically robust precursor-derived ceramics or PDC materials, such as silicon oxycarbide (SiOC). SiOC-functionalized TMDs have shown to curb capacity degradation in TMD and improve long term cycling as Li-ion battery (LIBs) electrodes. Herein, we report synthesis of such a composite in which MoSe_2_ nanosheets are in SiOC matrix in a self-standing fiber mat configuration. This was achieved via electrospinning of TMD nanosheets suspended in pre-ceramic polymer followed by high temperature pyrolysis. Morphology and chemical composition of synthesized material was established by use of electron microscopy and spectroscopic technique. When tested as LIB electrode, the SiOC/MoSe_2_ fiber mats showed improved cycling stability over neat MoSe_2_ and neat SiOC electrodes. The freestanding composite electrode delivered a high charge capacity of 586 mAh g^−1^_electrode_ with an initial coulombic efficiency of 58%. The composite electrode also showed good cycling stability over SiOC fiber mat electrode for over 100 cycles.

## 1. Introduction

The most recent advancement in energy storage and conversion systems was the instigation of flexible or bendable systems that were fabricated to be portable, lightweight, and wearable while being mechanically flexible and having a high-energy density [1]. However, the preparation methods, assembly process, and selection of proper electrolytes to transform traditional energy storage devices remains a challenging task. Conventional LIBs usually consist of a carbon-based anode, a transition metal oxide-based cathode, a polymer separator, and an organic liquid electrolyte. In such devices, heavy metal foils are always used as both a conductive substrate and structural support with a slurry of active materials, binders, and conductive additives coated on its surface. However, such electrode designs have certain drawbacks—for example, as the metal foil possesses a smooth surface, the active materials can easily become detached from the current collector. Additionally, there is an increase in the internal impedance and passivation of active materials due to relatively poor chemical stability. Lastly, the conductive additives such as carbon black, carbon nanotubes (CNTs), and graphite account for the 10–20 wt.% of the whole slurry make no or little contribution to the capacity of the cell. Polymer binders such as polyvinylidene fluoride (PVDF) are electrochemically inactive or even insulating, making their presence a hindrance to the transportation of lithium ions and electrons.

Flexible conducting and self-supporting electrodes replace traditional metal-foil-based heavy current collectors and standard binders with light-weight substrates or imbedded current collectors. The approach is to make a combination of active materials and flexible substrates by taking advantage of the conductive and flexible matrix embedded in the current collector or the substrates with functional materials firmly entrapped on them through novel methods such as coating, spraying, or depositing [2]. Although the use of CNT and graphene as both active materials and current collectors has been widely studied, other forms of carbon materials and some novel metal-based current collectors are now becoming more popular. For example, an effective chemical oxidation and reduction method to increase the surface area of carbon materials using a carbon cloth-based solid-state supercapacitor was prepared by Wang et al. [3]. A novel bamboo-like structure was reported in 2005, which showed excellent mechanical flexibility, foldability, and electrochemical performance. In dimethylformamide, polyacrylonitrile and tetraethyl orthosilicate were used to create nanofibers via electrospinning [4]. 

In recent years, the application of polymer-derived ceramic (PDC) materials and compounds with CNTs and graphene as free-standing electrodes in electrochemical devices has been studied. Due to desirable properties such as high stability during cyclic loading and structure preservation during very high C-rates, PDCs have become materials of increasing interest [5,6,7]. Interestingly, studies from David et al. [8] and Davi et al. [9] showed that the chemical functionalization of polymer-derived ceramics (PDC) with Transition Metal Dichalcogenide (TMD) materials leads to stable Li^+^ and Na^+^ cycling in the fabricated TMD-electrodes. Specifically, the presence of PDC nanolayers on TMD surfaces prevented the loss of sulfur species from the TMD surfaces during a conversion-type reaction.

In the present work, our primary motivation is to exploit the chemical functionalization of TMDs with PDCs and to demonstrate the fabrication of a new kind of freestanding electrode consisting of exfoliated MoSe_2_ TMD that offers Li ions stable cyclability, high energy density, and rate capability. This was achieved via a single-step electrospinning process in which a combination of liquid-phase organic/inorganic preceramic polymer and exfoliated TMD nanosheets were used as the spinning solution to fabricate non-woven fiber mats followed by pyrolysis to achieve a composite fiber mat of MoSe_2_ embedded in a SiOC matrix. The microscopy and spectroscopy images confirmed presence of MoSe_2_ in the SiOC matrix, and electrochemical tests of the SiOC/MoSe_2_ electrode in the LIBs showed enhanced performance over neat SiOC and bulk MoSe_2_ electrodes. To the best of our knowledge, a combination of the TMD MoSe_2_ and PDC in a free-standing form has not been reported for either battery or supercapacitor applications.

## 2. Materials and Methods

### 2.1. Materials

The preceramic silicon oligomer 1,3,5,7-tetramethyl, 1,3,5,7-tetravinyl cyclotetrasiloxane (denoted as 4-TTCS) was purchased from Gelest (Morrisville, PA, USA). Polyvinylpyrrolidone (PVP), which has an average molecular mass of 1,300,000 g mol^−1^, was used as the spinning agent, and dicumyl peroxide (DCP) was the cross-linking agent. Both materials were purchased from Sigma-Aldrich^TM^ (Missouri, MO, USA). Isopropyl alcohol (IPA), purchased from Fisher Chemical (Lenexa, KS, USA), was used as the solvent. This study also utilized ultra-high-purity and high-purity argon gas in the glove box and tube furnace, respectively. The gases were purchased from Matheson (Manhattan, KS, USA). Bulk molybdenum diselenide (99.9% pure) powder (MoSe_2_) was purchased from Alfa Aesar (Haverhill, MA, USA) and had a dark grey appearance with a molecular weight of 253.87.

### 2.2. Methods

#### 2.2.1. Electrospinning

The PDC fiber fabrication process begins by spinning the fiber generated from the preceramic polymeric precursor in sol–gel form. However, the raw material needed to produce the fibers must be a liquid and must have flowability or viscoelasticity and solubility in certain solvents to create an undistinguishable solution that is desirable for spinning and producing fibers. As for example, monomers and oligomers with infrequent repeating units tend to be liquid at ambient temperatures. Because the preceramic silicon polymer in this study possessed a low molecular weight (<500 g mol^−1^) and appeared to be a water-like liquid, linear polymers based on carbon materials with a high molecular weight were introduced to ensure that the sol–gel that was obtained would be viscous enough (>10 poise) for electrospinning.

Various parameters affect the fibers obtained via the electrospinning method. Among them, the following seemed to affect the fiber qualities:A stepper motor controls the speed of the polymer–solvent feed rate through the syringe. When the syringe capacity was fixed, the feed rate was directly affected by the rotating speed of the stepper motor. Thus, the high feed rate increased the distance from the needle tip to the collector or the voltage to be larger since the polymer solution had to be converted to fibers. Conversely, after continuous testing, a slower feeding rate ensured the production of good quality fibers with a lower applied voltage.The optimization of the applied voltage (15–20 kV) between the needle tip and collector ensured the formation of the fibers from the solution. However, a higher than optimal voltage resulted in the deposition of fibers in surrounding areas rather than in the collector.The needle tip and collector distance are adjustable depending on the feed rate of the polymer solution and applied voltage. The preferred space was 15–20 cm for the proper Taylor cone formation and for the creation of exceptional fibers without beads.Air bubbles in the polymer solvents resulted in an inconsistent feed rate through the syringe. Therefore, any air bubbles formed must be removed to ensure the continuous feeding of the solution.

Based on the above engineering parameters, the best conditions for fiber formation were determined on a trial basis and were implemented to produce fiber mats.

#### 2.2.2. Preparation of Fiber Mats

(a)PVP Fiber mat Preparation

In the solution preparation step for the PVP fiber mat, 800 mg of PVP powder was mixed with IPA to create a solution, and the ratio of PVP:IPA was kept at 1:9. A magnetic stirrer was used to create a homogenous mixture of the solution, which was stirred for 2 h. The process is demonstrated in Appendix A. The resulting solution was loaded into a 10 mL syringe with a flat metallic needle for electrospinning purposes. The fiber mat that was produced with this solution is referred to as the PVP fiber mat in the following sections.

(b)PVP/SiOC Fiber mat Preparation

The solution preparation for the PVP/SiOC fiber mat involved creating a similar solution as the one for the PVP fiber mat followed by the addition of a polymeric precursor material. The polymeric precursor 4-TTCS (i.e., 2400 mg) was primarily mixed with 1 wt.% DCP (i.e., 24 mg), which worked as a cross-linking initiator. The resultant PVP + 4-TTCS + IPA solution was stirred until a homogeneous mixture was achieved. The solution preparation process is illustrated in Appendix A. Finally, the above-mentioned solution was loaded in a 10 mL syringe with a flat metallic needle for electrospinning, and the fiber mat produced with this solution is referred to as the PVP/SiOC fiber mat in the following sections.

(c)PVP/SiOC/MoSe_2_ Fiber mat Preparation

The first step for the preparation of the PVP/SiOC/MoSe_2_ fiber mat involved the preparation of the nanosheet material from the bulk MoSe_2_ powder. For this particular purpose, 10 wt.% MoSe_2_ bulk powder compared to the amount of PVP (i.e., 80 mg) was added with IPA and was sonicated using a probe sonicator. In the next step, this solution was mixed with the same amounts of PVP and 4-TTCS as before and was stirred until a homogenous mixture was obtained. This solution preparation process is depicted in Figure 1a. The resultant solution was further loaded into a syringe and was electrospun to the obtain PVP/SiOC/MoSe_2_ fiber mat, as shown in Figure 1b. The feed rate for the fiber mat fabrication process was 5 mL h^−^^1^. An alligator pin was connected to the metallic needle and was subject to a high voltage of 15–20 kV depending on the viscosity of the solution, while the ground was connected to a cylindrical-shaped roller collector. The distance between the needle tip and the roller was kept at 15 cm. Thus, as-spun fiber mats that were 20 × 20 cm^2^ in size were fabricated. After drying in the open air, the as-spun fiber mats were put into a low-temperature oven and cross-linked at 160 °C for 24 h. The cross-linked fiber mats were cut into small pieces, were fitted into small rectangular (30 × 25 mm^2^) ceramic boats and put into a tube furnace in argon gas environment for high temperature pyrolysis. Two-step annealing was carried out at 400 °C for 1 h and at 800 °C for 30 min. The heating rate was kept at 2 °C per min. Thus, SiOC-functionalized MoSe_2_ fiber mats were produced. In addition, one sample of PVP/SiOC/MoSe_2_ fiber mat was fabricated by drop coating exfoliated MoSe_2_ (using the same amount of MoSe_2_ as before) onto a PVP/SiOC fiber mat followed by pyrolysis. This drop-coated PVP/SiOC/MoSe_2_ sample was prepared to compare its electrochemical performance with the electrospun PVP/SiOC/MoSe_2_ sample in which MoSe_2_ was inherently introduced to SiOC in a single-step spinning process.

#### 2.2.3. Electrochemistry

A 7.94 mm circular punch was used to punch out the composite pyrolyzed electrodes, which were then used as cathodes in the coin-like lithium half-cell setup. The weight of the PVP, PVP/SiOC, and PVP/SiOC/MoSe_2_ electrodes were 1.6 mg, 1.1 mg, and 2.1 mg, respectively. Pure lithium metal (diameter of 14.3 mm and a thickness of 75) µm was used as the counter electrode. The electrolyte solution was a mixture of 1 M lithium hexafluorophosphate (LiPF_6_) in 1:1 *v*/*v* dimethyl carbonate with ethylene carbonate (Sigma Aldrich^TM^, St. Louis, MO, USA). This solution had an ionic conductivity of 10.7 mS cm^−^^1^. A glass separator with a diameter of 19 mm and a thickness of 25 µm separated the two electrodes as they were pre-soaked with the electrolyte. The electrochemical measurements were taken using a multichannel BT 2000 Arbin test unit (College Station, TX, USA) between 2.5 V and 10 mV at current densities of 50, 100, 200, 400, 600, and 800 mA g^−^^1^.

### 2.3. Characterization Techniques

Scanning electron microscopy (SEM) was used to illustrate the morphology of the fiber mat samples. The SEM used was a Carl Zeiss EVO MA10 (White Plains, NY, USA) system with a 5–30 kV impinging voltage. Transmission electron microscopy (TEM) images of the samples were obtained using Phillips CM100 TEM (Nashville, TN, USA) under an accelerating voltage of 100 kV. The X-ray photoemission spectroscopy (XPS) spectra were obtained with a PHI Quantera SXM (Chanhassen, MN, USA) using monochromatic Al-Ka with an energy of 1486.6 eV to analyze the chemical composition of the surface. Raman spectra were collected using a He-Ne laser (wavelength of 633 nm and power of 17 mW) on a confocal Raman imaging system (Horiba Jobin Yvon LabRam Aramis, Edison, NJ, USA). Fourier transform infrared spectroscopy (FTIR) spectra of the fiber mats were obtained using the Spectrum 400 FT-IR spectrometer, PerkinElmer (Waltham, MA, USA).

## 3. Results and Discussion

### 3.1. Morphological Characterization

Because the fiber mats were first electrospun green and then cross-linked and pyrolyzed at elevated temperatures, the changes in the morphology of these fiber mats were investigated using SEM. Figure 2a–c illustrate the fiber-like morphology retention of the PVP, PVP/SiOC, and PVP/SiOC/MoSe_2_ fiber mats after the cross-linking process at 160 °C and pyrolysis at 800 °C. Figure 2d–f reveal the images taken for the pyrolyzed fiber mats at a higher magnification from which the average fiber diameters were calculated using a distribution plot, as shown in the inset of the respective images. From the distribution plots, the mean fiber diameters for the PVP, PVP/SiOC, and PVP/SiOC/MoSe_2_ fibers were identified as approx. 1.2 to 1.8 µm, approx. 1.2 to 2.75 µm, and approx. 2.25 to 3.75 µm, respectively. Compared to results in the literature, Wang et al. found that the average diameter of the PVDF/PVP nanofibers used in their experiment was in the range of 121–305 nm [10]. Additionally, the nanofibers fabricated by Newsome et al., which contained 9, 10, and 11 wt.% PVP in the silica/PVP electrospinning solution, possessed an average diameter in the range of 380–430 nm [11]. However, because the solution used and the electrospinning parameters differed from those used in the current research, the values cannot be directly compared. Ren et al. found that the average fiber diameter of the 4-TTCS fibers was 1 to 3 µm, which falls within the diameter range reported here [12]. Additionally, some irregularities such as microbeads and particles (likely to be MoSe_2_ in case of the SiOC/MoSe_2_ fiber mat) can be observed in the SEM images of the PVP/SiOC and PVP/SiOC/MoSe_2_ fiber mats. It can be inferred from the above information that the addition of polymeric precursor 4-TTCS turned into a silicon oxycarbide (SiOC) ceramic material, and the sonicated nanosheets composed of the TMD material (MoSe_2_) in the linear polymeric PVP fibers (transformed to carbon during pyrolysis) increased the diameters of the fibers sequentially. Thus, the presence of these materials in the fibers can be primarily confirmed and will be further validated by the other characterization techniques described in the following sections.

Figure 3a–c illustrates the low-mag TEM images of the PVP, PVP/SiOC, and PVP/SiOC/MoSe_2_ fibers carbonized or pyrolyzed at 800 °C, respectively, while the high-resolution TEM (HRTEM) images are shown in Figure 3d–f. From the lower and higher magnification TEM images of the fibers, it can be inferred that all of the fibers have dense cores, as the electrons could not pass through the fibers in the TEM instrument. The diameters of the pyrolyzed PVP, PVP/SiOC, and PVP/SiOC/MoSe_2_ fibers correspond well with the diameters in the SEM image, which validated the experiment. The carbonized PVP fibers seem to be uniform in diameter in both the higher and lower magnification images. Although mostly uniform along the cross-section, higher magnification images of the pyrolyzed PVP/SiOC and PVP/SiOC/MoSe_2_ fibers reveal the porosity and irregularity of the fibers. The beads seen from the SEM images of the PVP/SiOC and PVP/SiOC/MoSe_2_ fibers were thus further validated by the anomalies observed in the TEM images of the fibers. The fiber porosity was as expected because it is this porosity that enhances the use of such fiber mats as electrodes in energy storage devices. The irregularities and beads in these fibers can be attributed to the processing conditions of the fiber mats. During the cross-linking and pyrolysis steps, the long-chain polymeric compound PVP decomposes and forms the long chain.

The release of volatile compounds from the preceramic polymer resin during cross-linking and pyrolysis leads to irregular geometries similar to those observed in the investigation by Mujib et al. [13]. The non-regular features of the fibers can also be ascribed to solution parameters such as the molecular mass of the polymers used, correlating to the concentration of the polymer solution and viscosity and the effect of the solvent used, which has been widely studied in the literature [14,15,16,17]. Figure 3g–i represent the diffraction patterns of the pyrolyzed PVP, SiOC, and SiOC/MoSe_2_ fibers taken in the position previously seen in Figure 3d–f, respectively. Figure 3g,h, respectively, reveal the amorphous structure of the carbonized PVP and PVP/SiOC fibers, which correspond well with the investigation by Ren et al. [12]. The diffraction pattern of Figure 3i elucidates the crystalline nature of MoSe_2_ seen in the pyrolyzed PVP/SiOC/MoSe_2_ fiber mats by possessing an interplanar spacing of ~2.9 Å, which corresponds to (100) planes with six diffraction spots [18,19].

### 3.2. Spectrocopic Analysis

Raman spectroscopy is a fast and non-destructive ubiquitous method that was used to detect the microstructural features of the pyrolyzed fiber mats. The microstructural information of the fiber mats, including the free carbon phase and the presence of the TMD materials in the fiber mat, was determined using this versatile tool. Figure 4a,b,d elucidate the presence of the free carbon phase in the peak-fitted Raman spectra of the carbonized PVP fiber mat and pyrolyzed PVP/SiOC fiber mat and a definite region of the pyrolyzed PVP/SiOC/MoSe_2_ fiber mat, respectively. The convoluted plots above revealed the D band for all of fiber mats at approximately 1336 cm^−^^1^, which appeared because of the disorganization or disorder in the carbon phase [20]. The G band was observed in the region of approximately 1602 cm^−^^1^ due to the in-plane bond stretching of the sp^2^ carbons. Additionally, peaks were also observed at 1213 cm^−^^1^ and 1495 cm^−^^1^, which can be ascribed to the T and D” bands, respectively. While the T band can be ascribed to the presence of sp^2^-sp^3^ (C-C and C=C) bonds, the D” band corresponds to the presence of amorphous carbon in the sample. These two bands usually originate from graphene edges, from pores in the samples, and from sp^3^ hybridized carbon atoms. Riedel and co-workers found similar results with the micro-Raman spectroscopy of SiOC materials [21]. The Raman spectra obtained from other studies also indicate that the D and G bands are in similar positions [22,23,24]. Figure 4c illustrates the Raman spectra in the region of 200–320 cm^−^^1^ for both the bulk MoSe_2_ powder and pyrolyzed PVP/SiOC/MoSe_2_ fiber mats. For the bulk MoSe_2_ powder, two peaks appear at 239.29 cm^−^^1^ and 282.14 cm^−^^1^, among which the former shows a distinct sharp peak, and the latter has a dull one, indicating the A_1g_ and E^1^_2g_ modes, respectively. While the A_1g_ mode signifies the out-of-plane vibrations of the chalcogen (selenium) species, the E^1^_2g_ mode demonstrates the in-plane vibration mode of the transition metal (molybdenum) and chalcogen (selenium) atoms. In Raman-active A_1g_ mode, the Mo atom stays stationary, while the subsequent Se atoms that are present in all layers vibrate in phase with the subsequent Mo atoms [25]. The spectra obtained from the pyrolyzed PVP/SiOC/MoSe_2_ fiber mat in the region mentioned earlier shows A_1g_ and E^1^_2g_ modes that have two similar peaks in the slightly shifted region of 236.94 cm^−^^1^ and 287.03 cm^−^^1^, respectively. The slight shift to the lower wavelength is usually observed because of the softening of the A_1g_ mode (redshift), with a decrease in the thickness of the material [25]. While the A_1g_ peak was much more intense than the E^1^_2g_ peak in the bulk sample, the same peaks on the MoSe_2_ present in the PVP/SiOC/MoSe_2_ were found as broadened ones, indicative of the few-layered structure of the material [26,27]. Similar peaks in the same position have been observed in several other studies as well [28,29,30]. The intensity ratio of the D and G peaks from the plots for pyrolyzed PVP, PVP/SiOC, and PVP/SiOC/MoSe_2_ fiber mats were 1.02, 1.54, and 1.11, respectively, which revealed a higher degree of disorder-induced carbons in all of the samples.

Figure 5 illustrates the XPS survey scan of the carbonized/pyrolyzed PVP, PVP/SiOC, and PVP/SiOC/MoSe_2_ fiber mats. C 1s, O 1s, and N 1s peaks were visible in the survey scan of the carbonized PVP fiber mat, which validated the presence of these molecules, even after the pyrolysis stage. The survey scan of the pyrolyzed PVP/SiOC fiber mat revealed the presence of Si 2p, O 1s, and C 1s. For the pyrolyzed PVP/SiOC/MoSe_2_ fiber mat, two additional peaks, Mo 3d and Se 3d, were observed, which again substantiates the presence of MoSe_2_ in the pyrolyzed fiber mat. Table 1 presents the elemental composition of the pyrolyzed fiber mats, which was determined by integrating the area under the respective elemental peaks. The PVP fiber mat showed the highest amount of carbon present in the pyrolyzed stage, with a percentage of 93.06%. Some nitrogen was also found in the fiber mat (2.1%), which originated from the pyrrolidinone structure of the PVP. The elemental analysis of the PVP/SiOC fiber mats shows the presence of carbon, silicone, and oxygen. The high amount of carbon (70.46%) visible in the pyrolyzed fiber mat resembles the data obtained from the Raman analysis well. The presence of high amounts of oxygen (17.36%) in this fiber mat is likely due to the cross-linking process conducted in the presence of air. The surface analysis of the pyrolyzed PVP/SiOC/MoSe_2_ fiber mat reveals the presence of molybdenum and selenium particles in the fiber mat. The comparatively low amount of such particles only points to the elements present in the surface of the fiber mat or the fibers but not in the whole fiber mat. The nitrogen found in the pyrolyzed PVP/SiOC and PVP/SiOC/MoSe_2_ fiber mats was low compared to the PVP fiber mat, which did not contribute significantly to the total elemental analysis.

Figure 6 depicts the high-resolution scans of the XPS spectra for each element, elucidating the individual peaks obtained from the survey scan. Multiple peak fittings of the high-resolution XPS spectra reveal the elements being differently bonded with other elements (hybridization). The high-resolution N 1s spectra (Figure 6a) of the pyrolyzed PVP fiber mat disclose the presence of pyridinic, pyrrolic, graphitic, and oxidized nitrogen with binding energies of 398.2 ± 0.1 eV, 399.4 ± 0.1 eV, 400.8 ± 0.1 eV, and 402.7 ± 0.1 eV, respectively, which correspond well with the literature [31]. The peak fitting of the O 1s high-resolution spectra (Figure 6b) of the carbonized PVP fiber mat divulged the bonding states C-O, C=O, NO, and O-C=O at 535 ± 0.1 eV, 533.5 ± 0.1 eV, 532.5 ± 0.1 eV, and 531.5 ± 0.1 eV, respectively [32,33]. Additionally, the deconvolution of the C 1s high-resolution spectra (Figure 6c) uncovered peaks in the binding energy region of 287.5 ± 0.1 eV, 285.8 ± 0.1 eV, and 284.5 ± 0.1 eV, which can be ascribed to the bonding states of O=C-N, C-N, and C-C, respectively [34]. From the deconvoluted C 1s spectra of the pyrolyzed PVP/SiOC and PVP/SiOC/MoSe_2_ fiber mats, the C=O, C-C, and C-Si bonding states were observed in the regions of 287.1 ± 0.1 eV, 284.9 ± 0.1 eV, and 284.0 ± 0.1 eV, respectively. From the Si 2p spectra of the PVP/SiOC (Figure 6d) and PVP/SiOC/MoSe_2_ (Figure 6g) fiber mats, the SiCO_3_ and SiC_2_O_2_ bonding states were discovered in the 103.5 ± 0.1 eV and 102.9 ± 0.1 eV regions of the spectra. Finally, the O-Si and SiO_2_ bonding states were discerned from the O 1s spectra of the PVP/SiOC (Figure 6e) and PVP/SiOC/MoSe_2_ (Figure 6h) pyrolyzed fiber mats. The corresponding peaks align well with previous studies conducted regarding SiOC-based materials [12,35,36,37]. The high-resolution spectra also imparted the dominance of C-Si bonds over the C-C bonds. Furthermore, the deconvolution of the Mo 3d spectra of the pyrolyzed PVP/SiOC/MoSe_2_ fiber mats presented the 3d_5/2_ and 3d_3/2_ peaks in the region 228 ± 0.1 eV and, 231.1 ± 0.1 eV, respectively, signifying the 2H polymorph of MoSe_2_ [38,39]. In addition to this, two other peaks at around 229.8 ± 0.1 eV and 232.9 ± 0.1 eV were identified and were thought to have originated from 2H-MoSe_2_ [40]. Another peak at around 235.4 ± 0.1 eV was located and was determined to correspond to Mo-O-C bonds with low intensity values [41]. The spectra of the Se 3d spectra can be deconvoluted into two peaks of 3d_5/2_ and 3d_3/2_ in the regions of 54.1 ± 0.1 eV and 55.0 ± 0.1 eV for the 1T and 2H phases, respectively [42,43,44,45].

Figure 7a–c illustrates the FTIR absorption bands of the PVP, PVP/SiOC, and PVP/SiOC/MoSe_2_ fiber mats in their as spun, cross-linked, and pyrolyzed forms. From the FTIR spectra of the as-spun PVP fiber mat, absorption bands at 570, 841, 1018, 1288, 1422, 1663, 2923, and 3399 cm^−1^ were observed. These peaks can be assigned to N-C=O bond bending, CH_2_ bond bending, the rocking motion of CH_2_ bonds, C-N bond stretching, C-H bond stretching, C=O bond stretching, and symmetric stretching of the CH_2_ chain, respectively [46,47,48]. In the cross-linked fiber mats, similar peaks were observed at 160 °C, demonstrating meager changes indicative of the small cross-linking reactions of the polymer chains. Finally, from the bottom of Figure 7c in the PVP fiber mat carbonized at 800 °C, the peaks mentioned above were not visible, which indicates the presence of only carbon. From Figure 7a, the FTIR spectra in the middle is for the PVP/SiOC as-spun fiber mats from which the dedicated absorption bands for SiOC materials can be observed [12]. The peak observed at 797 cm^−1^ can be ascribed to the Si-C deformation vibration originating from the Si-CH_3_ bond. The peak at 1059 cm^−1^ is indicative of the Si-O-Si asymmetric stretching. Additionally, the peak at 1260 and 1423 cm^−1^ originated from the C-H symmetric and asymmetric bending vibration from the Si-CH_3_ bonds, respectively. The peak at 1662 cm^−1^ is thought originate from C=C double bond stretching. The absorption bands over the 2900 cm^−1^ can be allotted to the C-H asymmetric stretching originating from Si-CH_3_ and Si-CH=CH_2_ bonds. All of these bonds are in accordance with the previous literature and are suggestive of the presence of pre-ceramic polymers present in the as-spun fiber mats [12,49,50,51,52]. No significant changes were evident from the cross-linked fiber mat. However, the pyrolyzed fiber mat only represented the Si-O and Si-C bonds present in the 1060 and 800 cm^−1^ regions, which confirms the progressive degradation of the SiOC network as the temperature increases. Similar peaks were observed in the as spun, cross-linked and pyrolyzed PVP/SiOC/MoSe_2_ fiber mats and showed no additional peaks for MoSe_2_.

### 3.3. Electrochemical Analysis

Figure 8a–d illustrates the electrochemical charge–discharge behavior of the carbonized PVP, pyrolyzed PVP/SiOC, MoSe_2_ drop-coated pyrolyzed PVP/SiOC, and pyrolyzed PVP/SiOC/MoSe_2_ electrodes in the lithium half-cell setup over a voltage window of 0.01–2.5 V. In addition, Figure 8e–h denotes the differential curves of the same fiber mats in the previously mentioned order, where the plateaus of the charge–discharge profile are clearly visible with the corresponding peaks. From the charge–discharge behavior of the carbonized PVP fiber mat electrode presented in Figure 8a, for the first cycle, which took place at a current density of 50 mA g^−1^, a specific charge capacity of 569.96 mAh g^−1^ and a specific discharge capacity of 269.95 mAh g^−1^ are observed. Thus, high irreversible capacity loss can be observed, which is not visible in the consecutive cycles. It is possible to analyze this behavior with the help of the differential capacity curve from Figure 8e, where a sharp peak at 0.53 V is seen in the first cycle. However, this peak was not present in the consecutive cycles, which is indicative of an irreversible layer of materials, generally called solid electrolyte interface (SEI) forming, on the interface of the electrolyte and electrode. Similar peaks in can be observed in the 0.75–0.85 V region in various studies of hard carbons, which can be assigned to pore lithiation as well as SEI formation [53]. The peak below the 0.5V region can be attributed to the lithium ions interacting with the functional oxygen groups that are present in the fiber mat [54]. This peak was also irreversible in the delithiation cycle because the working voltage window does not fall into the range [55]. The peaks observed at around the ~1 V region originated from the nitrogen-containing carbon present in the fiber mat, which was confirmed by the XPS analysis [56,57].

Figure 8b,f shows the charge–discharge profiles and the differential capacity curves for the pyrolyzed PVP/SiOC fiber mat electrode. Fukui et al. demonstrated that the lithium ion storage mechanism of the SiOC material is via the intercalation/deintercalation reaction, which is quite similar to hard carbon materials [20]. The peaks seen in between 0.5–1.5 V can be assigned to the electrolyte decomposition and the formation of the SEI layer. In addition to these peaks, some other peaks can be observed in a low-lying potential region, which can be assigned to lithium being irreversibly trapped in a region where oxygen is present or where dangling bonds and defects are present. However, the delithiation curve obtained in Figure 8b is uniform and straight, showing neither any plateaus nor any peaks, which is indicative of the porous nature of the SiOC material [58,59,60,61].

Figure 8c,g shows the charge–discharge profiles and differential capacity curves of MoSe_2_ that was drop coated onto the electrode of the SiOC fiber mat—which was prepared for comparison purposes with the electrospun SiOC/MoSe_2_ fiber mat. Due to similarities in the structure of MoSe_2_ and MoS_2_, they go through the similar initial lithium ion intercalation [62]. As the MoSe_2_ material was introduced by creating a coating on top of the fibers, the lithium ions were more exposed to it, and thus, the peaks related to the reaction with the MoSe_2_ material were more pronounced. For instance, the peak observed in the first cycle near the 1 V region can be ascribed to the irreversible crystal structure change in MoSe_2_ from trigonal prismatic 2H-MoSe_2_ to octahedral 2T-Li_x_MoSe_2_ [63]. The latter co-ordination is more stable, and the phase change was confirmed by previous studies [64]. The peak assignment is based on the following lithiation process:(1)$${\mathrm{MoSe}}_{2}+{\mathrm{xLi}}^{+}+{\mathrm{xe}}^{-}\to {\mathrm{Li}}_{x}{\mathrm{MoSe}}_{2}$$

The first cycle lithiation peak at approx. 0.7 V can thus be assigned to the following conversion process:(2)$${\mathrm{Li}}_{x}{\mathrm{MoSe}}_{2}+\left(4-x\right)\mathrm{Li}\to \mathrm{Mo}+2{\mathrm{Li}}_{2}\mathrm{Se}$$

These two equations can be combinedly expressed as follows, and the formula has been confirmed by previous studies [62,65]:(3)$${\mathrm{MoSe}}_{2}+4\mathrm{Li}\to \mathrm{Mo}+{\mathrm{Li}}_{2}\mathrm{Se}$$

The peaks below the 0.4 V potential region are indicative of SEI formation by the SiOC material. Furthermore, one more peak situated in the 0.47 V can be assigned to the SEI formation that occurs due to the MoSe_2_ material. Finally, the oxidation peak visible in the ~2.1 V region can be accredited to the following equation confirmed by Cui et al., which appears in consecutive cycles as well [66]:(4)$$\mathrm{Se}+2\mathrm{Li}\to {\mathrm{Li}}_{2}\mathrm{Se}$$

For the pyrolyzed PVP/SiOC/MoSe_2_ fiber mat electrode (Figure 8h), three reduction peaks in the potential region of 0.4–1 V can be similarly credited to the lithiation process happening due to MoSe_2_, while the peak below this region is due to the SEI formation happening because of the SiOC material. Interestingly, no delithiation or oxidation peaks were observed in the first and second cycles, which is indicative of the dominance of the SiOC material in the fiber mat.

Figure 8i shows the comparison of the three fiber mat electrodes of the PVP PVP/SiOC mats and the PVP/SiOC/MoSe_2_ electrodes when they are cycled at increasingly higher current densities (100, 200, 400, 600, 800 mA g^−1^) in a rate capability test, thus leading to faster lithium ion insertion and extraction. The MoSe_2_ drop-coated electrode is not included in this comparison. The comparison of the first cycle charge capacity of the fiber mat electrodes show a dominance of the pyrolyzed PVP/SiOC/MoSe_2_ electrode, with a value of 586.96 mAh g^−1^ over the other electrodes. At higher the current densities at which the electrodes were tested, the PVP/SiOC fiber mat electrode showed a clear capacity advantage over the carbonized PVP fiber mat electrode. The reason for this good performance can be attributed to the porous nature of the SiOC materials and the fiber mats, which facilitate lithium ion storage by decreasing the distance required for ion diffusion when the electrolytes penetrate inside the deep pores. These pores also work as protection against the large volume expansion that takes place during the insertion and extraction of lithium ions [67,68]. From the investigation of Fukui et al., it can be confirmed that the SiOC materials store lithium ion on interstitial spaces or graphene edges [69]. Further improvements can be seen at even higher current densities for the pyrolyzed PVP/SiOC/MoSe_2_ electrode. Although the charge capacity of the pyrolyzed PVP/SiOC/MoSe_2_ electrode was slightly less than the PVP/SiOC electrode when cycled back to the initial 50 mA g^−1^ current density from 800 mA g^−1^, the fast recovery was achieved in the 71st cycle, and the specific charge capacity improved in the later cycles while cycling at the same current density.

The coulombic efficiency of the lithium half-cell for all of the electrodes is presented in Figure 8i. From figure, it is evident that all of the fiber mat electrodes suffered from a low coulombic efficiency in the first cycle, which is due to the formation of irreversible SEI formation. It is also evident that the pyrolyzed PVP/SiOC/MoSe_2_ fiber mat electrode was more stable while charging and discharging at the higher current densities, which indicates that the fast and reversible insertion and extraction of lithium ions into the fibers was maintained in all of the cycles. The pyrolyzed PVP/SiOC/MoSe_2_ electrode showed a stable response with a coulombic efficiency of ~100% for 100 cycles, whereas carbonized PVP and pyrolyzed PVP/SiOC electrodes showed poor reversibility when the current density was increased to 100 and 200 mA g^−1^. This further elucidates that the SiOC-functionalized MoSe_2_ fiber mat kept its structure intact, even after going through harsh testing conditions, making this fiber mat electrode superior to the other electrodes tested. A comparison of PVP/SiOC/MoSe_2_ electrode performance with previously studied SiOC electrodes in LIBs is included in Appendix A.

## 4. Summary and Conclusions

In summary, this work is the first to present MoSe_2_ nano-sheets combined with polymer-derived PVP/SiOC fiber mats to yield a free-standing electrode, providing stable cycling in higher current densities in LIB half-cell configuration. The ceramic mats were prepared via the electrospinning of chemically exfoliated MoSe_2_ that had been dispersed in preceramic polymer solution. A small quantity of PVP was used as a spinning agent, mainly to facilitate fiber spinnability. In addition to this, the evolution of the fibers from as spun to cross-linked and finally to the pyrolyzed form was confirmed by the FTIR spectroscopy. Raman and XPS spectroscopic studies revealed the presence of the free carbon phase present in the fibers and the mixture of bonds between Si-O-C, respectively. The PVP/SiOC/MoSe_2_ fiber mat electrode showed stable coulombic efficiency and a higher capacity than the fiber mats fabricated without the MoSe_2_ material when cycled at progressively increasing current densities. Therefore, a self-supporting SiOC-functionalized MoSe_2_ composite electrode possessing such stable performance during harsh cycling conditions can be suitable for a wide range of energy storage applications.

## Figures and Tables

**Figure 1 nanomaterials-12-00553-f001:**
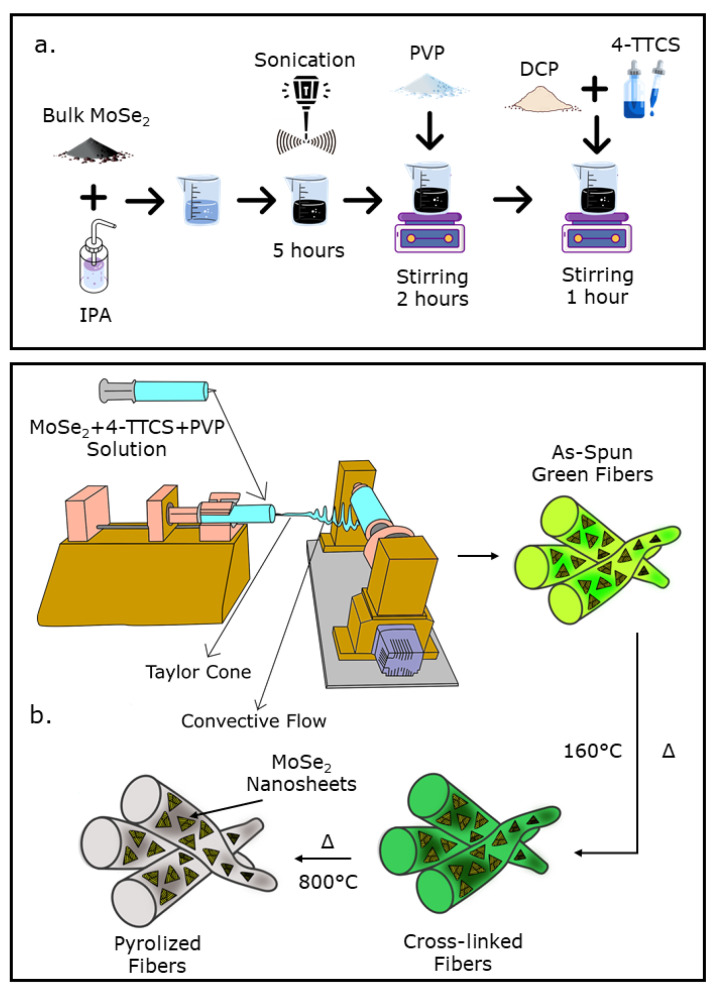
Solution preparation process for (**a**) PVP/SiOC/MoSe_2_ fiber mat; (**b**) schematic showing electrospinning process for the fabrication of the PVP/SiOC/MoSe_2_ composite fiber mats.

**Figure 2 nanomaterials-12-00553-f002:**
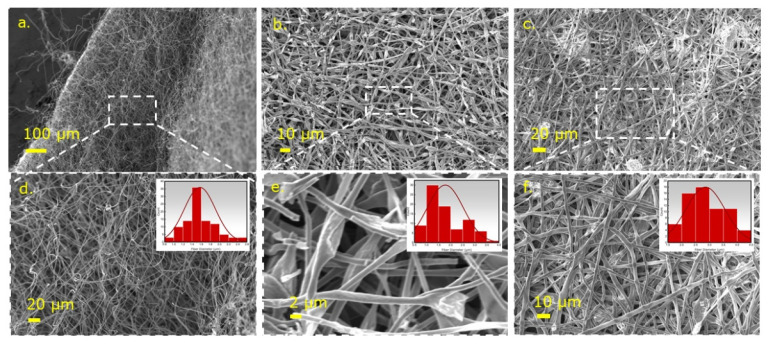
SEM images of (**a**) PVP fiber mat carbonized at 800 °C, (**b**) PVP/SiOC fiber mat pyrolyzed at 8000 °C, (**c**) PVP/SiOC/MoSe_2_ fiber mat pyrolyzed at 8000 °C, and (**d**–**f**) corresponding high-magnification images of these fiber mats (inset shows diameter distribution of the fibers).

**Figure 3 nanomaterials-12-00553-f003:**
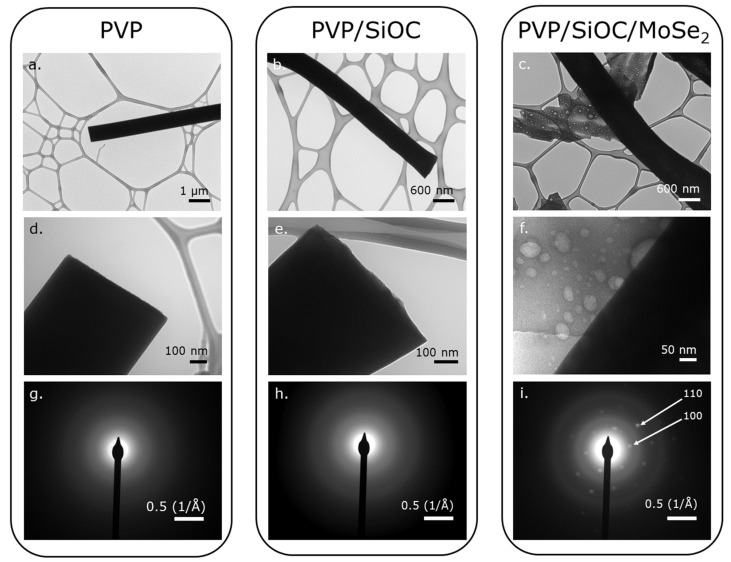
TEM images of (**a**) PVP fibers carbonized at 800 °C, (**b**) PVP/SiOC fibers pyrolyzed at 800 °C, (**c**) PVP/SiOC/MoSe_2_ fibers pyrolyzed at 800 °C, and corresponding HRTEM (**d**–**f**) and diffraction pattern (**g**–**i**) images of these fibers. Diffraction pattern of pyrolyzed PVP/SiOC/MoSe_2_ indicates the presence of MoSe_2_ particles in the pyrolyzed fiber mat.

**Figure 4 nanomaterials-12-00553-f004:**
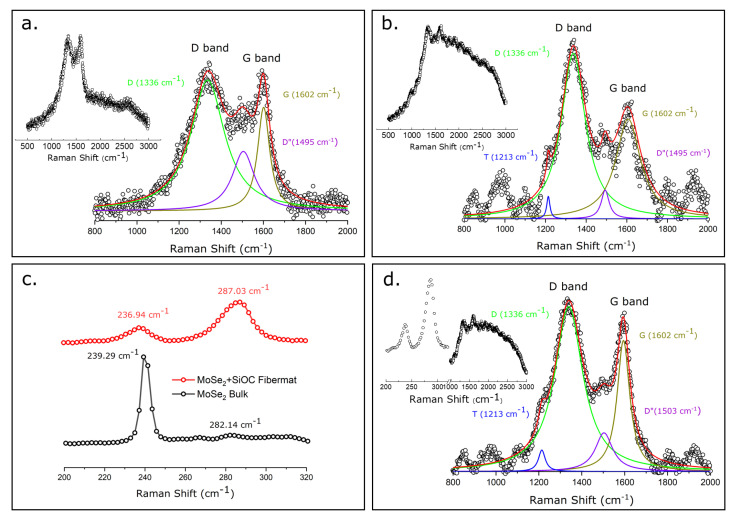
(**a**) Raman spectra of carbonized PVP fiber mat with integrated peaks; (**b**) Raman spectra of pyrolyzed PVP/SiOC fiber mat with integrated peaks; (**c**) Raman spectra of bulk MoSe_2_ powder and the region of 200–300 cm^−1^ from the pyrolyzed SiOC/MoSe_2_ fiber mat; (**d**) Raman Spectra of pyrolyzed PVP/SiOC/MoSe_2_ fiber mat with integrated peaks in the region of 800–2000 cm^−1^.

**Figure 5 nanomaterials-12-00553-f005:**
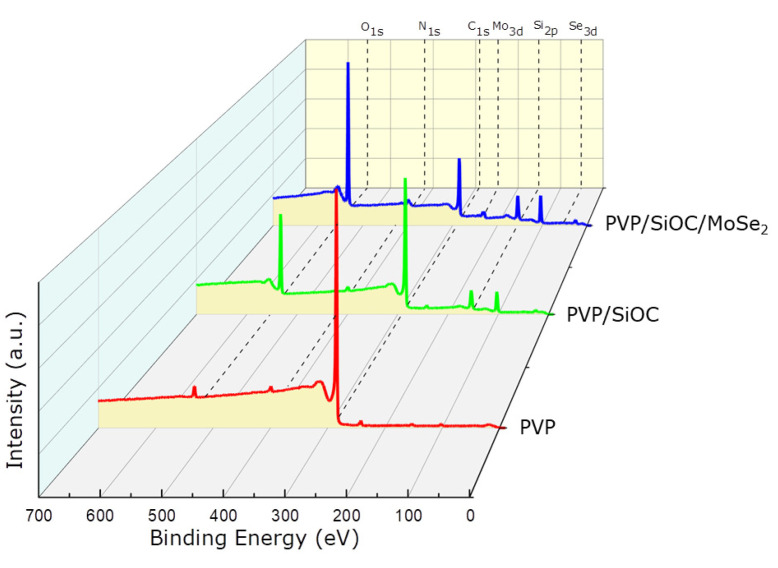
XPS survey scan of pyrolyzed PVP, PVP/SiOC, and PVP/SiOC/MoSe_2_ fiber mats.

**Figure 6 nanomaterials-12-00553-f006:**
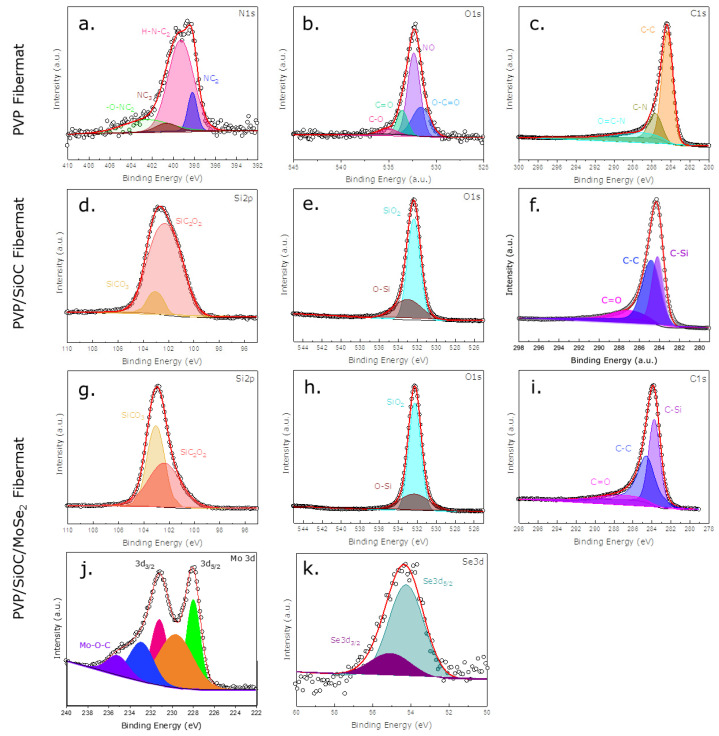
High-resolution XPS spectra of three pyrolyzed fiber mats: (**a**–**c**) PVP fiber mat; (**d**–**f**) PVP/SiOC fiber mat; (**g**–**k**) PVP/SiOC/MoSe_2_ fiber mat.

**Figure 7 nanomaterials-12-00553-f007:**
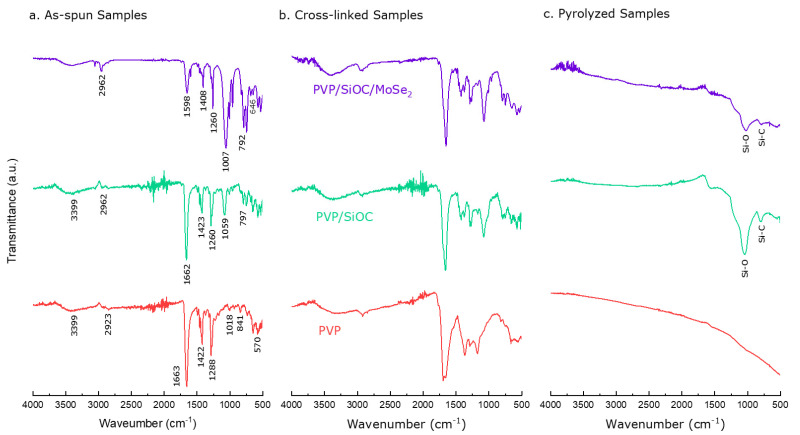
FTIR spectra of (**a**) as-spun, (**b**) cross-linked, and (**c**) pyrolyzed fiber mats. Each figure consists of the FTIR spectra shown for PVP, PVP/SiOC and PVP/SiOC/MoSe_2_ fiber mats from bottom to top.

**Figure 8 nanomaterials-12-00553-f008:**
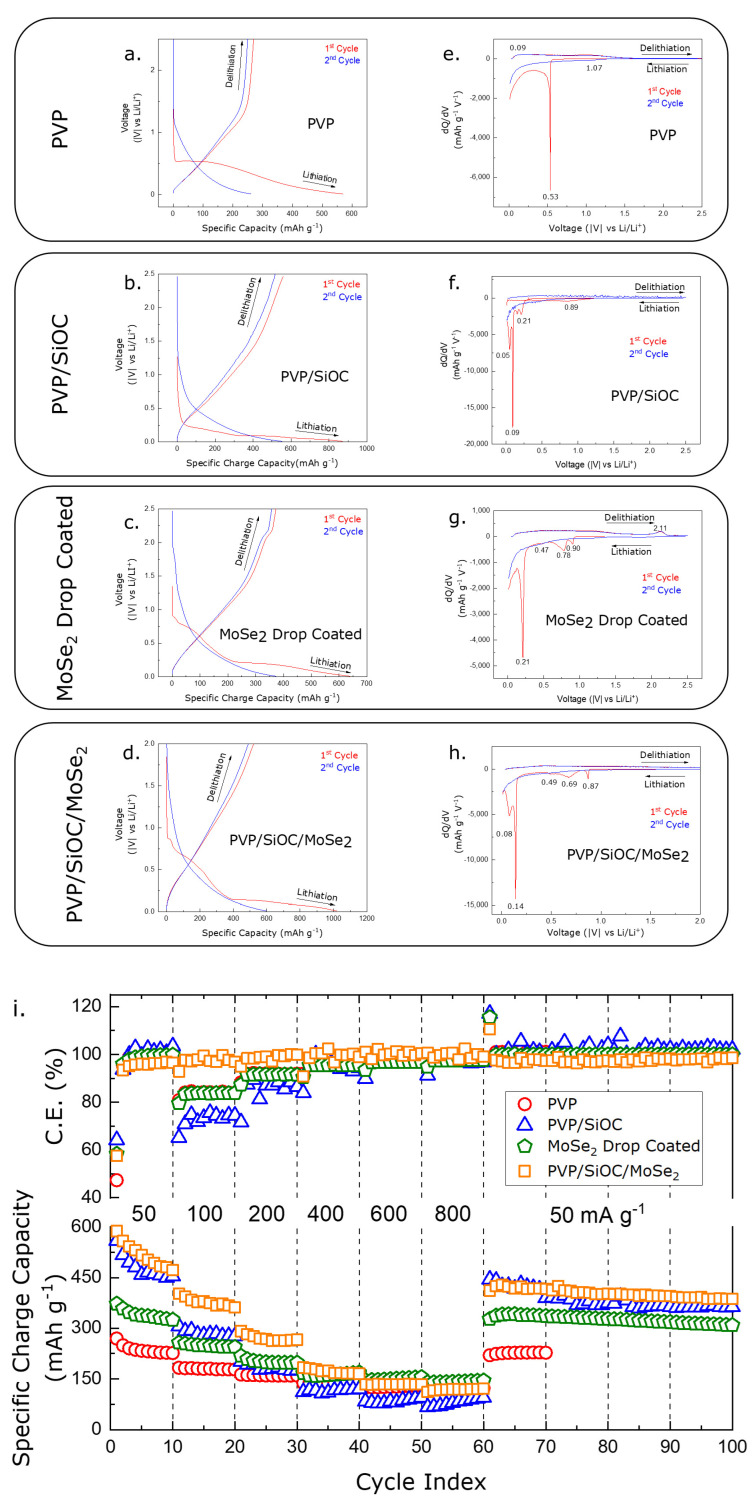
Charge–discharge profile of (**a**) PVP fiber mat electrode; (**b**) PVP/SiOC fiber mat electrode; (**c**) MoSe_2_ drop coated fiber mat electrode; (**d**) PVP/SiOC/MoSe_2_ electrode and differential capacity curves of (**e**) PVP fiber mat electrode; (**f**) PVP/SiOC fiber mat electrode; (**g**) MoSe_2_ drop coated fiber mat electrode; (**h**) PVP/SiOC/MoSe_2_ electrode; (**i**) cycle performance comparison for PVP, PVP/SiOC, PVP/SiOC/MoSe_2_ electrode at different current densities.

**Table 1 nanomaterials-12-00553-t001:** Elemental composition of the fiber mats by XPS.

Pyrolyzed Fiber Mats	Elements (Atomic %)					
	C	O	Si	N	Mo	Se
PVP	93.06	4.84	-	2.1	-	-
PVP/SiOC	70.46	17.36	12.18	-	-	-
PVP/SiOC/MoSe_2_	43.26	35.6	20.16	-	0.71	0.28

## Data Availability

The data presented in this study are available at “https://ksuemailprod-my.sharepoint.com/:f:/g/personal/sbmujib_ksu_edu/EgORagImlxFGuSz06YgxAggBbwQvHqHWO_kKWqIMwazFkg?e=e28Lv1 (accessed on 27 December 2021)”.

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
