# Peer review of "Enhanced Li-Ion Rate Capability and Stable Efficiency Enabled by MoSe2 Nanosheets in Polymer-Derived Silicon Oxycarbide Fiber Electrodes"

_nanomaterials, 2022, doi:10.3390/nano12030553_

Round 1
Reviewer 1 Report
Dear authors,
Please see my comments below.
Lines 63-66: The syntax of this sentence is a bit confusing. Please correct it.
Fig.1: Improvement, more details needed in figure and caption. Larger pictures and schematics. Possibly separate the electrospinning format (fig. 1a) and the synthesis details (fig.1b), include the details for the other samples (shown at the characterisation and electochemistry part), ie. PVP fiber mat, PVP/SiOC fiber mat pyrolyzed at 800 oC, PVP/SiOC/MoSe2 fiber mat pyrolyzed at 800oC, drop coated MoSe2. Might be an option to include a table with the samples and the preparation conditions, so it is easy to follow. Include experimental details for PVP fiber mat and PVP/SiOC fiber mats at the experimental section.
Lines 92-93: Could include details regarding the MoSe2 powder
Lines 129-149: Please provide mass, volume and/or concentration of the precursors used for the synthesis.
Lines 147-149: drop coated SiOC/MoSe2 sample vs electrospun SiOC/MoSe2 sample: would this be PVP/SiOC/MoSe2 ?
Line 150: Please correct the paragraph title. It should be Electrochemistry
Lines 151-152: What was the weight of the electrodes? Could you comment on the amount of MoSe2 in the samples used for the electrochemistry?
Fig.2 (SEM): It should be 800 oC (not 8000C). Insets showing the diameter distribution are too small. Provide images of higher resolution. Also add the scale bars.
Fig.4 (Raman): Insets are too small. Include details in caption. Fig. 4D, seems to be a good indication for MoSe2 in the final product PVP/SiOC/MoSe2, which is compared to bulk MoSe2 at fig. 4C. Although the positions of the peaks seem similar, there is a big difference at the intensities, which could be due to morphology differences? Could discuss at lines 256-260 and compare with literature.
Lines 297-299: Please move Table 1 after Fig.5.
Table 1. Please explain why no Se is detected in the sample PVP/SiOC/MoSe2
Fig. 6 d, g : Please use different color than yellow. Please explain why you don’t use thew orbital splitting peaks for Si2p as you did for Mo3d and Se3d.
Fig.7 (FTIR): Could rearrange the spectra and categorize them A. as spun, B. Cross-linked, C. Pyrorolysed. At the current form the differences are not necessarily visible to the reader. Provide higher resolution images.
Fig. 8 (a-h): Should declutter so it is easier for the reader to follow. The images are too small and the resolution is poor. Would it be easy to include all the four samples in only two subfigures (1st and 2nd cycle), with the same colour code as in fig. 8I-cycle performance? Should include the data for MoSe2 drop coated in fig. 8I and if you have data for MoSe2 powder.
Please correct typos in electrochemistry equations- should include literature reference.
????2 + ???+ + ??− → ???????2
???????2 + (4 − ?)?? → ?? + 2??2?e
????2 + 4?? → ?? + 2 ??2??
Lines 444-449: Strong point of the paper. Should compare with literature- possibly include a table with data for MoSe2 fabricated/measured with different techniques. Include in the abstract.
Author Response
Thank you for your valuable time and effort. The revised version of the manuscript with track changes enabled, reviewer's comments with responses, supplementary section, and another version of the manuscript with changes highlighted have been uploaded.

Reviewer 2 Report
1-To enhance the capacity performance, the anode materials need to be high conductivity in both electron and lithium ions. Why the Si-based composite could enhance the rate performance of MoSe2, as the silicon oxycarbide mentioned by authors may not have higher ionic conductivity that that of carbon.
2-In the TEM images, you can show which one is MoSe2 as you mentioned <... presence of MoSe2 particles in the pyrolyzed fiber mat>. Also, the rings in the diffraction patterns should be marked correctly.
3- I have one question on this. The in-plane and out-of-plane of MoSe2 indicates the orientations, which is the vertical or parallel. Please check of this and give some deep explanations.
4- In the XPS spectrum of Mo-3d, some fitting should be given for the distinguish of valence states.
5- For the electrochemical performance, the mass loading of MoSe2 should be given. Besides, we did not know the capacity contributions of Si-based materials and carbon in the SiOC/MoSe2 composites. Please clarified those information.
Author Response
Thank you for your valuable time and effort. Please see the file named reviewer's comment 2 attached.

Round 2
Reviewer 2 Report
The current manuscript could be published as is.